# In Situ Response Time Measurement of RTD Based on LCSR Method

**DOI:** 10.3390/s25154826

**Published:** 2025-08-06

**Authors:** Yanyong Song, Yi Liang, Zhenwen Zhang, Geyi Su, Mingxu Su

**Affiliations:** School of Energy and Power Engineering, University of Shanghai for Science and Technology, Shanghai 200093, China

**Keywords:** response time, resistance temperature detectors, loop current step response, plunge test

## Abstract

This study aims to overcome the limitations of conventional plunge tests for evaluating resistance temperature detector (RTD) response times under actual operating conditions, particularly in confined nuclear power plant piping. An in situ measurement device based on the loop current step response (LCSR) method was developed, with a conversion relationship to plunge test results established through numerical simulation and experimental validation. Investigations in a rotating water channel (over the flow velocity range of 0.2 to 0.6) confirmed excellent agreement in RTD response time, showing only 3.78% deviation between second-order-converted LCSR and plunge test measurements at 0.6 m/s. Both methods consistently revealed reduced RTD response times at higher flow velocities, with deviations consistently within ±10%, complying with nuclear instrumentation standards (NB/T 20069-2012). The LCSR method enables reliable in situ assessment while maintaining strong correlation with laboratory plunge tests.

## 1. Introduction

Resistance thermometers are widely employed across various industrial processes, owing to their advantages of high measurement accuracy, excellent sensitivity, long-term stability, and user-friendly operation without the need for cold-end compensation [1]. Particularly in nuclear power plants, the coolant temperature within the primary circuit piping serves as a direct and critical indicator of reactor power. Significant temperature transients require reliable, real-time delivery of temperature signals to reactor control and protection systems. Moreover, there are strict requirements on the dynamic performance of operating resistance thermometers. The response time of resistance temperature detectors (RTDs) is a key indicator of the dynamic performance of temperature measurement systems. The fast and effective response of RTDs in nuclear reactors ensures timely reactor shutdown during significant transient temperature fluctuations [2,3,4].

Response time, typically denoted by the symbol *τ*, is usually defined as the time it takes for the output signal to reach 63.2% of its stabilized value. Response time limits imposed by RTDs are typically outlined in the plant technical specification requirements and are enforced by regulatory regulations such as those in Regulatory Guide 1.118 and NUREG-0800 [5]. If the RTD fails to meet the specified response time requirements, unexpected transient temperatures can occur in the reactor coolant system. At this point, the safety of the nuclear power plant may be jeopardized, requiring the timely shutdown of the plant or other mitigation measures. Therefore, accurate measurement of RTD response time is critical and has attracted significant attention regarding applications of resistance thermometers in nuclear power plants.

According to the current standards of the nuclear power industry, the measurement of response time is mainly categorized into the plunge test and loop current step response (LCSR) test. The plunge test is used for offline measurement in laboratory conditions, where the RTD is quickly inserted into a high-temperature fluid. The external temperature change triggers a temperature step, which has been widely used and recognized. However, since the installation and process conditions of an RTD have an impact on the response time measurement, offline measurements after disassembly cannot accurately reflect its in-service response time. The LCSR method replaces the sudden change in the external thermal environment with the internal-step Joule heat to achieve in situ, online measurement of the resistance thermometer, avoiding damage to the sensor during disassembly or transportation and thus providing an objective assessment of the in-service temperature sensor accuracy, troubleshooting, and residual life. Therefore, the LCSR method shows a good application prospect in the field of nuclear power [6,7].

At present, the study of the factors influencing the response time based on the LCSR method of measurement is still difficult. During field testing, the flow velocity range of the external fluid is usually limited, while the fluctuation in the fluid temperature can cause perturbation to the LCSR transient process. Rupnik et al. [8] designed a dynamic characteristic measurement system for the LCSR method of RTDs based on virtual instrumentation technology and carried out performance tests with PT100 as the object. Cai et al. [9] investigated the effect of heating current, time, and number of tests on the response time measurement of RTDs in LCSR experiments. Klebba et al. [10] analyzed the factors affecting the response time from the point of view of heat transfer and optimized the calibration process of RTDs based on the response time. Bai et al. [11] optimized the temperature sensing element, housing structure, and relative airflow angle of air inlet holes through simulation analysis to improve the dynamic characteristics of RTDs. Zhang et al. [12] analyzed the influence of skeleton material and skeleton structure on the response time of RTDs based on FloEFD simulation software and designed a corresponding skeleton structure to carry out experimental verification. While the LCSR method has been extensively investigated, experimental studies directly comparing its performance with the plunge test under controlled flow conditions remain scarce. Crucially, the quantitative relationship between medium flow velocity and response time measurement deviation, a key factor for in situ sensor calibration, has yet to be systematically established. This necessitates further research to address the issue.

The present study aims to develop an in situ response time measurement device based on the LCSR. First, we detail how a numerical model based on cylindrical wall heat transfer theory was established, employing computational simulations to investigate the influence of external flow conditions on LCSR-derived response times, which also helped to establish and verify the conversion relationship between the LCSR method and the plunge test. Next, the principles, experimental procedures of the methods involved, and self-developed LCSR measurement device are presented. Finally, for test conditions in a rotating sink with flow velocities ranging from 0.2 to 0.6 m/s, experimental results are compared with those of the plunge test, and the influence of flow velocity on the RTD response time is analyzed, followed by discussions and conclusions.

## 2. Principles

Response time measurement methods can be generally categorized into the LCSR method and external thermal excitation (plunge test), based on the type of step signal generation [13,14,15,16]. The LCSR method is particularly suitable for in situ measurements of RTDs in process pipelines due to its adaptability, timeliness, and operational simplicity. Meanwhile, the plunge test is regarded as a widely accepted standard technique. Therefore, a comparative experimental study of these two methods—focusing on their principles, implementation procedures, and results—is of practical significance.

### 2.1. Response Time Measurement Principle of in Situ Method Based on LCSR

The LCSR method is an effective technique for in situ measurement of resistance thermometers [17,18,19]. In this method, a step current of several milliamperes (mA) is applied to the resistance leads at a specific moment. This current then generates Joule heat within the resistor, causing a temperature rise in the resistor itself and thereby facilitating the heat exchange with surrounding fluid at the interface. The resistor temperature stabilizes when the heat dissipated through the interface per unit time balances the Joule heat generated by the current. This process describes the transient temperature variation in the resistance thermometer induced by an internal current step.

The theoretical model of the resistance thermometer is derived from heat transfer equations. Analytical results indicate that the transfer function for the temperature change in the resistance thermometer (caused by variations in the external fluid temperature) can be expressed as [6](1)$$H_{1}\left(P\right)=\frac{K}{\Pi_{i}P\left(P-P_{i}\right)}$$

For the Joule effect producing self-heating leading to a change in the temperature of the resistance thermometer, the transfer function is(2)$$H_{2}(P)=\frac{\Pi_{i}(P-Z_{i})}{\Pi_{i}(P-P_{i})}$$
where *K* is a constant, *P* is the Laplace operator, $P_{i}$ represents the poles of an equation, and $Z_{i}$ denotes the zeroes.

The transfer functions given in Equations (1) and (2) are derived through inverse Laplace transformations of the resistance thermometer modal response equations, corresponding to a step change in external temperature and a step change in internal Joule heating, respectively. These equations are expressed as Equations (3) and (4):(3)$$R_{1}(t)=A_{0}+\sum A_{i}exp(-P_{i}t)$$(4)$$R_{2}\left(t\right)=B_{0}+\sum B_{i}\exp\left(-P_{i}t\right)$$
where *A*_0_, *A*_i_, *B*_0_, and *B*_i_ represent the constants in the equations obtained when converting the frequency domain transfer function to the time domain dynamic response equation.(5)$$A_{0}=\frac{1}{\left(-P_{1})\left(-P_{2}\right)\ldots (-P_{i}\right)}$$(6)$$A_{1}=\frac{1}{\left(P_{1}\right)\left(P_{1}-P_{2}\right)\ldots \left(P_{1}-P_{i}\right)}$$(7)$$A_{2}=\frac{1}{\left(P_{2})\left(P_{2}-P_{1}\right)\ldots (P_{2}-P_{i}\right)}$$

Thus, $A_{i}$ is determined by the pole values $P_{i}$ of the equation, with the reciprocals *τ_i_* of the latter referred to as modal time constants. Fundamentally, the LCSR and plunge test methods share identical governing equations and consequently exhibit identical pole configurations (subject only to differing boundary conditions). This mathematical equivalence establishes the theoretical foundation for their mutual conversion. For LCSR-based in situ response time measurement, the response time is derived from the model-predicted response curve. The conversion procedure is typically implemented as follows:Acquire the raw LCSR curve by applying an internal current step.Perform nonlinear fitting of the raw LCSR curve using Equation (4) to extract the *P_i_* values.Calculate *A_i_* by substituting the fitted values into Equations (5)–(7).Generate the modal response equation for external step changes by substituting *P_i_* and *A_i_* into Equation (3). Subsequently, predict the response curve through the model and determine the response time.

### 2.2. Principle of Plunge Test Response Time Measurement

For the plunge test, which typically involves rapidly immersing the sensor into a fluid with a higher or lower temperature, an external fluid temperature step can serve as a driving source. The transient response temperature of a resistance thermometer is continuously recorded from the initial to the final state to obtain a step response curve. The thermal response time is defined as the time required for the resistance thermometer to follow a step change in temperature and reach a specified percentage [20]. When reporting the response time, it is necessary to state the percentage of response. Typically, the dynamic characteristics of the RTD is defined as the time it takes for the output change to reach 63.2% of the total change in output when the RTD is excited by a step signal. Figure 1 shows a schematic diagram of the dynamic response process.

Based on the analysis of heat transfer, it is known that the response time of resistance thermometer is not only related to the structural shape, material, and heat transfer area of the thermometer itself but also closely related to the properties of the medium and the flow state. Therefore, to perform the comparable test, the surrounding medium and the flow state should be clearly provided.

## 3. Numerical Simulation

### 3.1. Heat Transfer Modeling

For simplification without loss of generality, the RTD response time test procedure can be considered as a one-dimensional unsteady state heat conduction problem for an infinitely long cylindrical wall. The governing equation is [21](8)$$\frac{\partial^{2}T}{\partial r^{2}}+\frac{1}{r}\frac{\partial T}{\partial r}=\frac{1}{\alpha }\frac{\partial T}{\partial t}\;a<r<b,t>0$$
where thermal diffusivity *α* = *λ*/*ρCp*, with the thermal conductivity *λ*, density *ρ*, and specific heat capacity *Cp.* The boundary conditions are as follows:

LCSR:(9)$$\frac{\partial T}{\partial r}=-\frac{Q}{\lambda }\;r=a$$(10)$$h\left(T-T_{f}\right)=\lambda \frac{\partial T}{\partial r}\;r=b$$

Plunge test:(11)$$\frac{\partial T}{\partial r}=0\;r=a$$(12)$$h\left(T-T_{f}\right)=\lambda \frac{\partial T}{\partial r}\;r=b$$

The initial conditions are as follows:(13)$$T=T_{0}=T_{f}\;t=0$$
where *T* is the temperature inside the cylinder wall with inner and outer radii a and b, respectively, as a function of the time *t* and radius *r*. The initial temperatures of the cylinder wall and the external fluid are *T*_0_ and *T*_f_, respectively. *Q* is the heat source applied to the inner wall surface, and *h* is the convective heat transfer coefficient, which apparently reflects the effect of fluid flow on the heat transfer process. The relationship between the convective heat transfer coefficient *h* and fluid flow velocity *u* can be found in Equation (14) [22].(14)Nu=0.3+0.62Re1/2Pr1/31+0.4Pr2/31/4×1+Re2820005/84/5
where the Nusselt number (*Nu* = 2*hb*/*λ*) represents the dimensionless temperature gradient of the fluid at the wall surface, the Reynolds number (*Re* = 2*ub*/*v*, with freestream velocity *u* and kinematic viscosity *v*) denotes the ratio of fluid inertial forces to viscous forces, the Prandtl number (*Pr* = *v*/2*b*) indicates the ratio of momentum diffusivity to thermal diffusivity.

The heat transfer process is analyzed using the lumped parameter method, a simplified approach for analyzing transient heat transfer when the Biot number is small (typically Bi < 0.1) [22]. Node *i* varies in temperature, *T_i_*, across time and space. The transient heat transfer equation for node *i* can be expressed in terms of the mass m of the nodal material, specific heat capacity *Cp*, heat source *Q_i_*, and heat transfer thermal resistances *R_i_* and *R_i_*_−1_.(15)$${\left(mCp\right)}_{i}\frac{dT_{i}}{dt}=\frac{1}{R_{i}}\left(T_{i+1}-T_{i}\right)+\frac{1}{R_{i-1}}\left(T_{i-1}-T_{i}\right)+Q_{i}$$
where *R_i_* is the thermal resistance between node *i* and node *i* + 1, *i* = [1, *n*], and *n* is the number of nodes. Clearly, Equation (15) provides a fundamental framework for the numerical analysis of heat transfer processes, enabling simultaneous evaluation of both the LCSR and plunge test methods to elucidate their conversion relationship (e.g., as defined in Equations (3) and (4)). However, careful consideration must be given to the modeling entities and boundary conditions involved. In a representative case study, the space step and time step were set to be 0.0001 s. The inner radius of the infinite-length cylinder walls *a* = 1 mm and the outer radius *b* = 3 mm, with a linear gird interval of 0.1 mm. For the plunge test, a sensor initially at 20 °C was rapidly immersed into 100 °C water. For the LCSR method, both the RTD and ambient environment started uniformly at 20 °C, with the heat source defined by a step function (e.g., as defined in Equation (16)) that instantaneously increased to *Q* = 3574.14 W/m^2^ (for *t* ≥ 0). The remaining parameters are shown in Table 1.(16)$$Q=\frac{I^{2}R}{\pi DH}$$

Here, *I* represents the step current, consistent with the experimental setup at 50 mA. *R* denotes the resistance of the platinum sensing element at 20 °C, measured as 107.79 Ω. *D* and *H* correspond to the diameter and height of the support substrate, measuring 1.6 mm and 15 mm, respectively.

As the critical external parameter, fluid velocity dictates RTD response time through its deterministic control of the interfacial heat transfer coefficient (Equation (14)), which governs the sensor thermal response dynamics. Fundamentally, high fluid velocities can enhance the heat transfer of the thin film on the sensor surface and generate a reduced response time. As shown in Figure 2, for a given flow velocity, the wall temperature gradually increases with time, converging to the fluid temperature. While, the convective heat transfer coefficient increases gradually as the flow velocity increases from 0.2 m/s to 0.6 m/s, the final equilibrium temperature reached gradually decreases and the time for the temperature profile to reach balance gradually decreases.

### 3.2. Numerical Simulation and Verification

Figure 3 presents the dynamic response curves of the LCSR method and the plunge test across different flow velocities. It should be noted that due to the differing temperature increases between the LCSR method and the plunge test, the normalization of data was employed. In principle, this normalization process did not affect the response time and enabled a more effective comparison of the dynamic response curve variation characteristics. With the increase in flow velocity, the slope of the dynamic response curve gradually becomes larger and the response time gradually decreases. In the initial stage of the dynamic response of the plunge test, due to the thermal inertia of the resistance thermometer filler material, there was a short delay time in the temperature response of the resistance wire when the external temperature changed in steps. The LCSR method generated heat directly from the sensing element, and the response curve is much steeper in the initial phase and can follow external temperature changes more quickly. Due to the differences in the heat generation method and the direction of heat transfer, the calculated dynamic response curves show that the plunge test response times at different flow velocities were 1.18 s, 0.86 s, and 0.72 s, while the LCSR method response times were 0.99 s, 0.68 s, and 0.54 s, respectively. With the increase in flow velocity from 0.2 m/s to 0.6 m/s, the relative errors of the two response times were enlarged from 16.10% to 20.93% to 25.00%. It is clear that the relative error of both response times gradually became larger with the increase in flow velocity due to the presence of delay time.

In accordance with the procedure provided in Section 2.1, the nonlinear Equation (4) was used to fit the LCSR dynamic response curve to obtain the *p*_n_ value of its modal response equation. The iterative fitting process utilized the Levenberg–Marquardt optimization algorithm for solving nonlinear least squares problems. It combined the advantages of gradient descent and Gauss–Newton methods and adapted itself between them [24,25]. Then, the known *p*_n_ and *A*_n_ from Equation (5) to Equation (7) were substituted into Equation (3) to obtain the equivalent modal equations, and the response curves’ conversion from the LCSR method to the plunge test was implemented. Thus, a transformed response time could be calculated, corresponding to the RTDs’ output to reach 63.2% of its final stead-state value following an outer step change in the temperature of the surrounding fluid.

To further investigate the optimal conversion order of the LCSR method, the LCSR curves were converted according to the first and second orders (*n* = 1, 2) and compared with the response curves of the plunge test. Figure 4 shows that the dynamic response curves (0.66 s and 0.69 s) obtained by LCSR after conversion are closest to the plunge test (0.72 s), with relative errors of only 8.22% and 3.78% from the plunge test when *n* = 1 and 2. In addition, for the other two flow velocities, the maximum relative error was found to be 6.31% for low-order models.

## 4. Experiments

As shown in Figure 5a, the experimental setup consisted of five components: a rotating sink, a mechanical arm, a control system, a test cabinet, and a LabVIEW-based data acquisition and processing software system. The software system offers both data acquisition functionality for recording dynamic response curves and data analysis capability for analyzing response time characteristics. The entire experimental setup was developed in-house, including hardware and software systems. The experimental system features a PLC-controlled rotating water sink that maintains precise isothermal conditions (20–100 °C with ±0.1 °C stability) while generating adjustable flow velocities (0–1.5 m/s). Real-time temperature control is achieved through PID-regulated heating power adjustment based on feedback from immersed sensors, while servo motor-driven speed modulation enables accurate flow regime generation. This setup is primarily designed for laboratory environments, enabling precise characterization of response time under controlled hydrodynamic and thermal conditions, where the LCSR method and the plunge test were used to measure the response time of the platinum RTD. As shown in Figure 5b, an armored resistance thermometer was selected as the test sample (Zhejiang Lunte), in which the ceramic-skeleton platinum resistor served as temperature sensing element, with the permitted temperature measurement range (−200~600) °C and an accuracy level of A grade (overall diameter *D* = 6.4 mm).

In order to verify the validity and accuracy of the LCSR method and device, the response time test was performed and the results were compared under the same conditions as those in the plunge test, including the medium, flow velocity, and initial temperature of the RTD, respectively.

The plunge test required adjusting the motor speed and the heating power of the inner wall surface of the sink through the control system in advance to achieve a constant temperature and speed state. Then, the RTD was controlled to reach thermal equilibrium in the sink at 20 °C. Afterwards, the mechanical arm inserted the RTD into the sink at 60 °C to realize the step change in the external temperature of the RTD, and the response time *τ*_0.632_ was calculated. The test was repeated three times, taking the average of the three test results, with the deviation within ±10%. The response times were tested at three flow velocities separately.

The LCSR test required the RTD to be placed in the sink at 20 °C. A 1 mA excitation current was input to the RTD through the test cabinet to remove significant internal temperature gradients. After the signal remained stable, a 50 mA step current was input for testing. The transient temperature curve generated by the internal current step was recorded, and the response time *τ*_0.632_ was calculated by model fitting. Five tests were repeated, and the average value of the five test results was taken, with the deviation within ±10%. The response times at three flow velocities were tested separately.

Both experiments used an NI USB-6212 data acquisition card (16 bits, 400 kS/s) to obtain the dynamic response curve at a 1 kHz sampling rate. To ensure that the entire dynamic response of the RTD was recorded, the acquisition time was set to be more than five times the response time. Finally, the dynamic performance characteristics were further analyzed by the developed program. The program automatically identified the point in time when a step change in the response curve occurred. Meanwhile, the program conducted the algorithm of a nonlinear curve fitting as described in Section 3.2 to implement the conversion of the response curve from the LCSR method to the plunge test, thereby determining the response time.

## 5. Results and Discussion

### 5.1. Response Curves and Conversion

The LCSR and the plunge test curves are presented according to the raw response curves obtained using the internal current step and the external temperature step, respectively. To validate the LCSR test and establish its comparability with the plunge test, a theoretical conversion is fundamentally required. The model-predicted response curves (the converted LCSR curves), then, were calculated by the methods described in Section 2.1.

As shown in Figure 6, Figure 7 and Figure 8, the LCSR, model-predicted and plunge test curves at water flow velocities of 0.2 m/s, 0.4 m/s, and 0.6 m/s are plotted, respectively. According to the plots of the step response curves at different flow velocities, the internal current step results depict a steeper curve for the resistance change. This is because when there was a step current through the platinum sensing element, the temperature firstly increased due to the Joule effect, and temperature rise led to the temperature’s difference from that of the surrounding fluid, which consequently led to heat coming from the platinum resistance sensing element, through the filler conduction to the outer wall. Convective heat transfer could occur on the interface of the outer wall and the fluid, and progressively with time, the temperature difference became larger. When the Joule heat power and convective heat transfer were the same, thermal equilibrium could be reached again.

Since the heat was generated directly on the sensing element by the LCSR method, the step response curve is steeper in the initial stage compared with that for the plunge test where the heat was conducted to the sensing element through the outer wall, and the difference in the step response curves of the two methods reflects the difference in the mode of action of the heat and the direction of heat transfer.

Table 2 gives the response times and the deviation obtained from the LCSR curves, the model prediction, and the calculation of the plunge test results at different flow velocities; the maximum value of the deviation of the LCSR curves from the plunge test results is −83.41%, and the maximum value of the deviation of the model prediction results from the plunge test results is only −3.48%. This shows that the deviation between the raw LCSR and plunge test results is large; thus, the LCSR curve cannot be directly used to indicate the response time of the measured sample. Meanwhile, the model prediction using the in situ method based on the LCSR step data has a good consistency with the plunge test, meeting the requirements of the relevant standards, which are within ±10%, so it can be applied in the in situ measurement of platinum RTDs. It is worth emphasizing that although the RTD used in the experiment was sheathed (effectively constituting a multi-layer cylindrical wall structure), the established conversion relationship between the LCSR and the plunge test was successfully employed throughout the experimental procedure.

### 5.2. Effect of Flow Velocity on Response Time Test Results

The results of the response time and relative standard deviation of measurements calculated using the LCSR-based model prediction and plunge test at different flow velocities are given in Table 3 and Table 4. With the increase in flow velocity, the test values of the response time of both the LCSR-based model prediction and plunge test become smaller, and both have the same trend. The maximum values of the relative standard deviations of the LCSR-based model predictions and measurements using the plunge test are 1.47% and 1.49%, respectively, with similar dispersion of the experimental data.

### 5.3. Scope of Application of Methods

The response time of platinum RTDs is not a constant. The medium properties and flow velocity simultaneously affect the RTD response time, in which the negative correlation with flow velocity can clearly be found. Nuclear power plant standards require that the response time of RTDs without casing should be less than or equal to 3 s. With casing, the corresponding critical time is 20 s. In a steam pipeline with flow velocity up to 45 m/s, the technical specifications require fast-response RTDs with a casing response time of less than or equal to 8 s.

The LCSR-based response time measurement principle for platinum RTDs is widely applicable. However, two aspects of the influencing factors should be emphasized. Firstly, the step current should fall in the scope of the platinum RTD and be as high as possible, to make the original curve have a better signal-to-noise ratio. Secondly, the response time of the platinum RTD should be determined under the specific environmental conditions, and this response time should be within the permitted response time measurement range of the measuring device.

In addition, the hardware parameters of the measuring device are the main factors affecting the response time measurement. The sampling period and the sampling accuracy of the device should be optimized, to apply a smaller current step for a faster response time curve with higher time resolution, meeting the requirements of faster response time measurement at higher flow velocities.

## 6. Conclusions

An LCSR-based measurement device was developed to realize in situ response time measurement of resistance thermometers. The validity and applicability of the device was verified through comparative analysis with the plunge test in the laboratory conditions (after conversion), with a maximum relative error of 3.48% at different flow velocities. The following conclusions are drawn:(1)The effect of external fluid flow velocity on the LCSR time was analyzed by numerical simulation. Specifically, high fluid velocities increase the heat transfer coefficient of the thin film on the sensor surface and decrease the response time. The relative error between LCSR and plunge test response time is significantly reduced from 25.00% to 3.78% (at a flow velocity of 0.6 m/s) by the second-order conversion.(2)Under the condition of flow velocity (0.2–0.6) m/s, the model prediction based on the LCSR and the response time test results of the plunge test have good consistency, and the maximum value of the deviation of the model prediction results from the plunge test results is only 3.48%, which meets the requirements of the monitoring test standard for the performance of safety-important instrumentation channels in nuclear power plants (NB/T20069-2012).(3)The LCSR-based model prediction has the same trend of change in and dispersion of measurement results regarding the response time as that of the plunge test, which verifies the response time test of the resistance thermometer.(4)The response curves of the LCSR test are steeper than those of the plunge test, which reflect the differences in the transient changes in the resistance value of the sensing element due to different heat actions and directions of heat transfer, i.e., self-heating of the sensing element and the change in external temperature.

The LCSR method, with an internal current step, can be implemented in the laboratory and in situ conditions without disassembling the on-site instrumentation. It achieves accuracy comparably to the plunge test while offering high detection efficiency and wide adaptability. Our future work will focus on enhancing the in situ measurement accuracy and expanding the applicability of the LCSR method, while concurrently conducting comprehensive site adaptation studies for nuclear power plant applications.

## Figures and Tables

**Figure 1 sensors-25-04826-f001:**
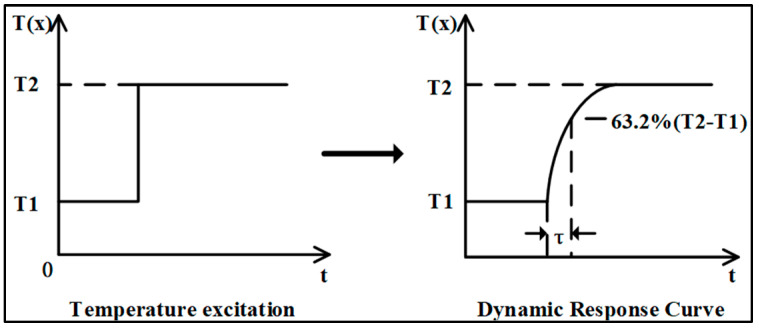
A schematic diagram of the dynamic response process.

**Figure 2 sensors-25-04826-f002:**
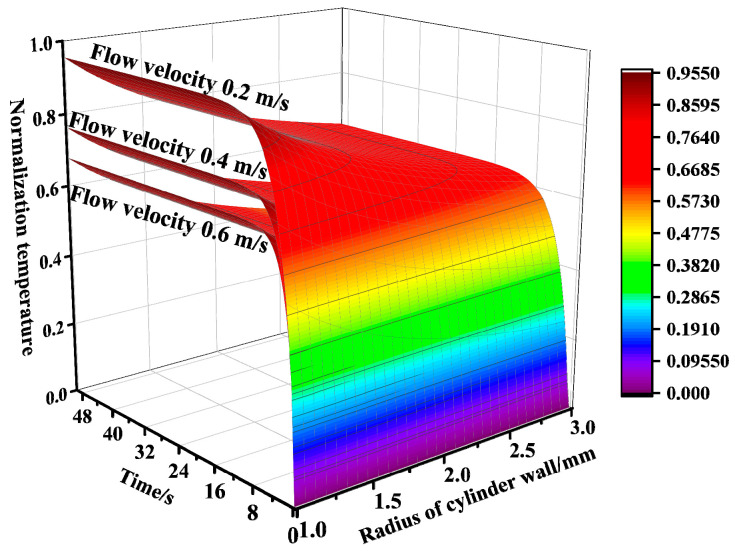
Temperature field distribution of RTD at different flow velocities.

**Figure 3 sensors-25-04826-f003:**
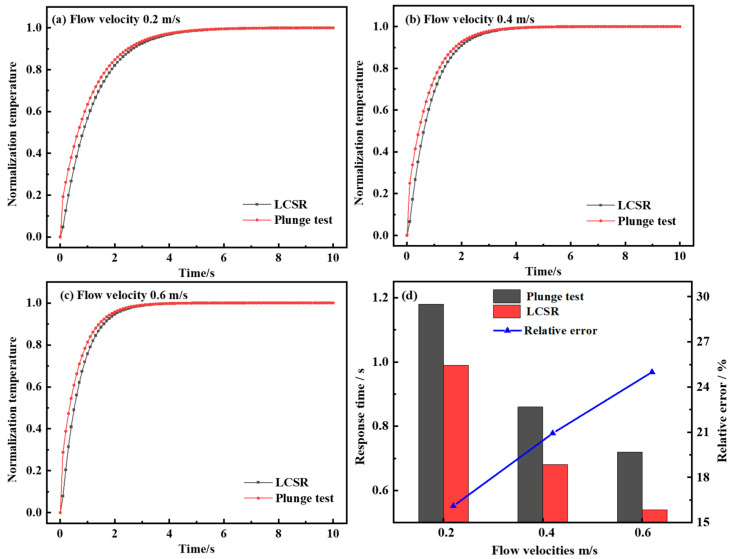
(**a**) Dynamic response curves of LCSR and plunge tests at flow velocity 0.2 m/s. (**b**) Dynamic response curves of LCSR and plunge tests at flow velocity 0.4 m/s. (**c**) Dynamic response curves of LCSR and plunge tests at flow velocity 0.6 m/s. (**d**) Comparison of response times and relative errors between LCSR and plunge tests across varying flow velocities.

**Figure 4 sensors-25-04826-f004:**
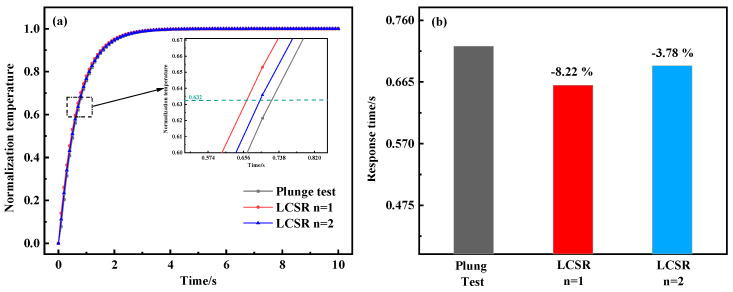
(**a**) Dynamic response curves from the plunge test and fitted LCSR method (*n* = 1, *n* = 2). (**b**) Response time and relative error of the plunge test and fitted LCSR method (flow velocity 0.6 m/s).

**Figure 5 sensors-25-04826-f005:**
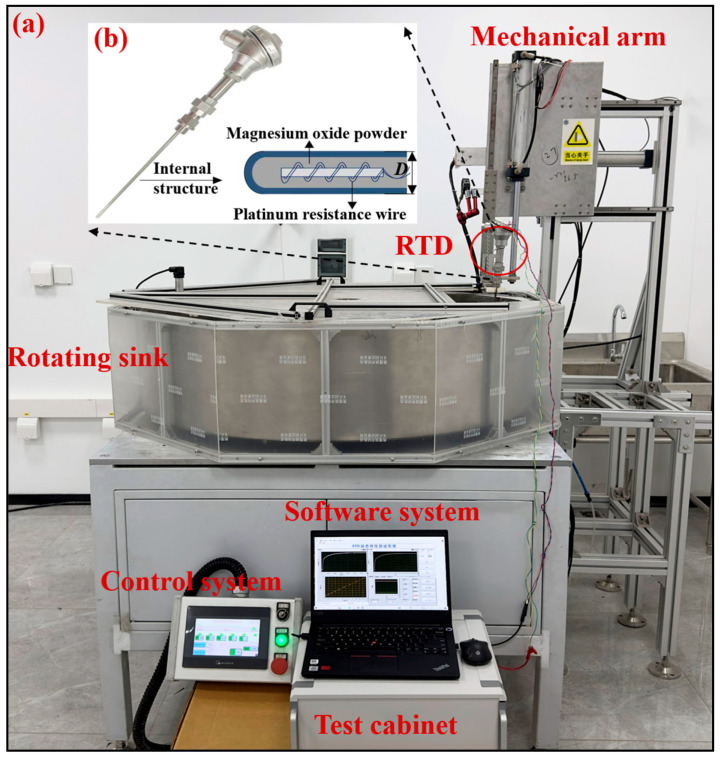
Experimental setup and test sample: (**a**) Five-component setup including rotating sink, mechanical arm, control system, test cabinet, and software system. (**b**) The armored resistance thermometer test sample featuring ceramic-skeleton platinum sensing element.

**Figure 6 sensors-25-04826-f006:**
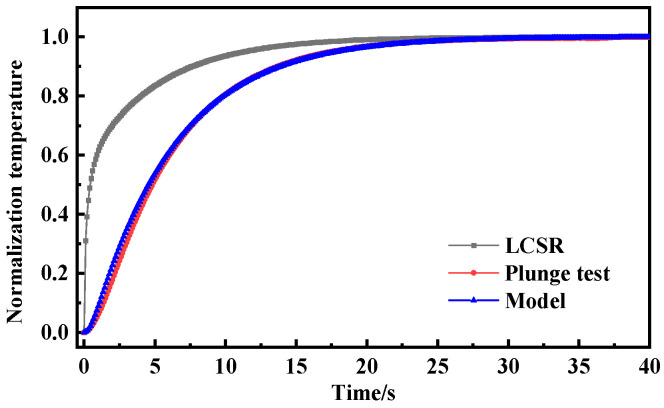
Step response curve of water flow velocity of 0.2 m/s.

**Figure 7 sensors-25-04826-f007:**
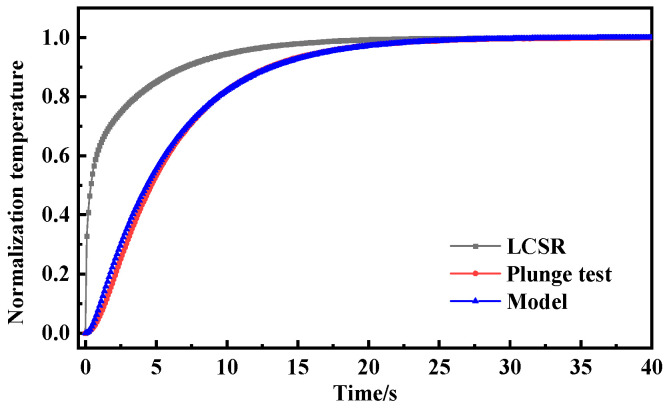
Step response curve of water flow velocity of 0.4 m/s.

**Figure 8 sensors-25-04826-f008:**
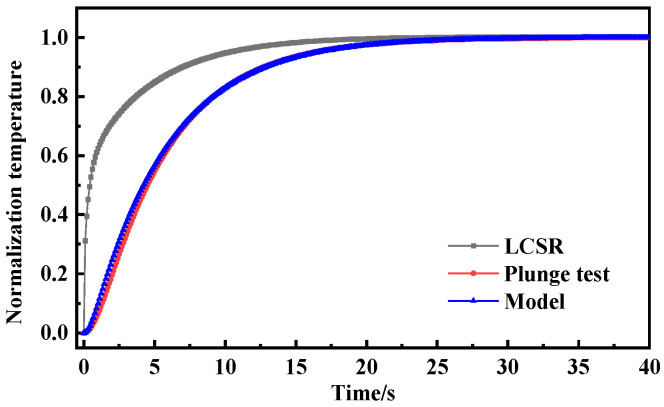
Step response curve of water flow velocity of 0.6 m/s.

**Table 1 sensors-25-04826-t001:** Material thermo-physical parameters of RTD [23].

Sensor Parts	Material	Density kg/m^3^	Constant Pressure Heat Capacity J/(kg·K)	Thermal Conductivity W/(m·K)
Filling material	Magnesium oxide	3648	920	43
Fluid media	Water	998	4187	0.59

**Table 2 sensors-25-04826-t002:** Measurement result of response time.

Item	Flow Velocity m·s^−1^	LCSR *τ*/s	Model *τ*/s	Plunge Test *τ*/s	LCSR and Plunge Test Deviation/%	Model and Plunge Test Deviation/%
1	0.2	1.12	6.50	6.49	−82.74	0.15
2	0.4	1.05	6.11	6.33	−83.41	−3.48
3	0.6	1.03	5.97	6.11	−83.14	−2.29

**Table 3 sensors-25-04826-t003:** Response time according to plunge test at different flow velocities.

Item	Flow Velocity m·s^−1^	Plunge Test τ/s			Average τ/s	Relative Standard Deviation/%
1	0.2	6.45	6.49	6.54	6.49	0.69
2	0.4	6.24	6.42	6.29	6.32	1.49
3	0.6	6.12	6.06	6.15	6.11	0.69

**Table 4 sensors-25-04826-t004:** Response time according to model prediction at different flow velocities.

Item	Flow Velocity m·s^−1^	Model Prediction *τ*/s					Average *τ*/s	Relative Standard Deviation /%
1	0.2	6.45	6.44	6.44	6.66	6.54	6.50	1.47
2	0.4	6.11	6.11	6.15	6.13	6.03	6.11	0.73
3	0.6	5.98	5.99	5.96	5.94	5.97	5.97	0.33

## Data Availability

The datasets generated during and/or analyzed during the current study are available from the corresponding author upon reasonable request.

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
