# Peer review of "In Situ Response Time Measurement of RTD Based on LCSR Method"

_sensors, 2025, doi:10.3390/s25154826_

Round 1
Reviewer 1 Report
Comments and Suggestions for Authors
In this work, the authors did a comparative study of the response time measurement of Pt RTDs, including the LCSR method and external thermal excitation (plunge test). The research topic is interesting. However, I can not recommend its publication in the present form.
- I strongly suggest the authors to polish abstract to be more concise.
- What are the behind mechanisms for the difference between LCSR and plunge test?
- In Fig. 7, it looks like the the simulation model matches well with the plunge test, whereas, LCSR exhibits more difference. Could the authors comment on this?
Author Response
Dear Editors and Reviewers:
Greetings! We would like to thank the reviewers for their insightful questions and comments, which are helpful to improve our manuscript significantly. We have revised and proofread the article in accordance with the review’s comments, and major changes in the revised manuscript are marked in red, which are explained below. In responding to the comments of different reviewers, we used four colors, orange, purple, blue, and green, corresponding to four reviewers. A more detailed explanation of the above can be found in the responses to reviewers below.

Reviewer 2 Report
Comments and Suggestions for Authors
Dear Authors,
I have read the submitted manuscript with great attention and interest. In my opinion, the research results are important and timely. I believe the manuscript is suitable for publication in Sensors, provided that the necessary revisions are made. Please revise the manuscript according to my comments.
- In my opinion, the Abstract is too long. I suggest significantly shortening it. Please clearly highlight the purpose of the work, explain what is new in the context of the state of the art, and summarize the main results obtained.
- Please ensure that all expansions of acronyms are written in lowercase, as in "resistance temperature detector (RTD)". The entire manuscript should be revised accordingly.
- Formulas (1), (7), (10), (12), and (16) contain incorrect punctuation at the end. Please revise the manuscript accordingly.
- Formulas (1) and (2) are written incorrectly. Please correct them and provide appropriate references.
- Lines (132)-(135) contain incorrectly formatted sentences. Please revise.
- I note that unnecessary indentations appear in several places in the manuscript, e.g. lines 140, 180, 185, 191, and 200. The manuscript should be revised, and the indentations should be removed.
- Sections 2.2 and 4 cannot begin with figures. Please revise your manuscript.
- Please provide a reference for formula (8).
- Line 169: The sentence "The heat transfer process is analyzed using the lumped parameter method" requires explanation. Please provide a reference for this sentence.
- Please provide a reference for the data presented in Table 1.
- Please check the unit names in Table 1.
- Line 273: In my opinion, the phrase "acquisition process" lacks precision. I suggest revising this part of the manuscript.
- Lines 307-308: Please provide more information in the manuscript regarding the sentence “Finally, the dynamic performance characteristics were further analyzed by the developed program.”
- Please revise the titles of Sections 3.2 and 5.1, as they require correction.
Yours sincerely,
Reviewer
Author Response

(The authors gave the same response as above.)

Reviewer 3 Report
Comments and Suggestions for Authors
1-Equation 2 on page 3. Line 134. Pj should be Pi
2-Experimental setup should be disclosed. Information such as manufacturer, model, catalog and part number should be mentioned? If any of the setup was built in house, then the authors should explain how they were built.
3-Figure 3 and 4: The figure caption should be more descriptive. For example, Figure 4a: Dynamic Response of normalization temperature versus Time…Figure 4b:…. The same goes for Figure 3a, 3b, 3c and 3d.
4- Figure 5 caption should be more detailed.
5-Conclusions (1) Line 386-400: concluding remarks should be more specific by mentioning numbers or percentages.
6-The authors should also mention what has been achieved. Have they met the objectives of their work?
7- The authors should mention whether there is any future work
Author Response

(The authors gave the same response as above.)

Reviewer 4 Report
Comments and Suggestions for Authors
Dear authors,
This is an interesting work allowing RTD sensor time constant estimation with in-situ measurements.
In Equ.1,, 2, 5,6,7 and 16 there is a prime ( ‘) in the denominator, what does it mean?
In Table 1, for the density please replace Kg/m^3 by kg/m^3 and for the heat capacity J/(Kg.K) by J/(kg.K) , to avoid any confusion with Kelvin
Could you specify the overall diameter of the platinum RTD you are working on? Unless I am mistaken, I did not find it in the manuscript.
I fully agree with your comments about the fact that time constant depends also strongly on the quality of the heat exchange between the sensors and its environment
Would it be possible to perform such work also with thermocouple? A current for ex. about 10mA can heat or cool down the hot junction of a thermocouple about a few degree depending on its diameter?
As a conclusion, I do think that this work is worth publishing. The manuscript is well written, discussion and conclusion are pertinent. I would recommend to accept this work for publication after (a few) minor corrections.
Best regards
Author Response

(The authors gave the same response as above.)
